# Enhancing Mechanical Properties of 3D Printing Metallic Lattice Structure Inspired by Bambusa Emeiensis

**DOI:** 10.3390/ma16072545

**Published:** 2023-03-23

**Authors:** Shikai Jing, Wei Li, Guanghao Ma, Xiaofei Cao, Le Zhang, Liu Fang, Jiaxu Meng, Yujie Shao, Biwen Shen, Changdong Zhang, Huimin Li, Zhishuai Wan, Dengbao Xiao

**Affiliations:** 1School of Mechanical Engineering, Beijing Institute of Technology, Beijing 100081, China; 2China Aerospace Science and Industry Corporation, Beijing 100081, China; 3State Key Laboratory of Mechanical Structure Strength and Vibration, School of Aerospace, Xi’an Jiaotong University, Xi’an 710049, China; 4Hubei Key Laboratory of Theory and Application of Advanced Material Mechanics, School of Science, Wuhan University of Technology, Wuhan 430070, China; 5The 41st Institute of Fourth Academy of Aerospace Science and Technology Corporation, Xi’an 710049, China; 6Xi’an Research Institute of Surveying and Mapping, Xi’an 710054, China; 7Beijing Key Laboratory of Lightweight Multi-Functional Composite Materials and Structures, Institute of Advanced Structure Technology, Beijing Institute of Technology, Beijing 100081, China; 8School of Mechanical Engineering, Nanjing University of Science and Technology, Nanjing 210094, China

**Keywords:** lattice structure, 3D printing, mechanical properties, Bambusa emeiensis, bionic design

## Abstract

Metallic additive manufacturing process parameters, such as inclination angle and minimum radius, impose constraints on the printable lattice cell configurations in complex components. As a result, their mechanical properties are usually lower than their design values. Meanwhile, due to unavoidable process constraints (e.g., additional support structure), engineering structures filled with various lattice cells usually fail to be printed or cannot achieve the designed mechanical performances. Optimizing the cell configuration and printing process are effective ways to solve these problems, but this is becoming more and more difficult and costly with the increasing demand for properties. Therefore, it is very important to redesign the existing printable lattice structures to improve their mechanical properties. In this paper, inspired by the macro- and meso-structures of bamboo, a bionic lattice structure was partitioned, and the cell rod had a radius gradient, similar to the macro-scale bamboo joint and meso-scale bamboo tube, respectively. Experimental and simulated results showed that this design can significantly enhance the mechanical properties without adding mass and changing the printable cell configuration. Finally, the compression and shear properties of the Bambusa-lattice structure were analyzed. Compared with the original scheme, the bamboo lattice structure design can improve the strength by 1.51 times (β=1.5). This proposed strategy offers an effective pathway to manipulate the mechanical properties of lattice structures simultaneously, which is useful for practical applications.

## 1. Introduction

Lightweight lattice structures with designable cell configurations and customizable properties have a wide range of application prospects in different fields, such as aerospace and medicine [1,2,3,4]. Geometric parameters, such as the cell topology and slenderness ratio, can change the mechanical properties of lattice structures. Thriving additive manufacturing (AM) technology [5] has enabled the design and fabrication of lattice cells with complex configurations. Nevertheless, metallic components filled with lattice cells always have a false geometric design or exhibit unsatisfactory mechanical performances caused by volumetric porosity, surface roughness, radius variation, strut waviness, and surface defects [6,7,8,9,10,11]. The former typically means that the designed model exists only in the drawing and is not suitable for a printing process. For example, Figure 1a depicts a multifunctional satellite cabin component with lattice cells and geometric constraints. When the stacking direction of the components is determined (Figure 1b,c), it is difficult to print lattice cells with an inclination angle θ between the cell rod and the stacking direction that exceeds 43°. The latter means that the mechanical properties of AM lattice structures will be reduced by 20–30% compared with the design value because of manufacturing defects, such as rod diameter fluctuations and waviness [9,10]. Generally, the cell configurations range from body-centered cubic (BCC) to face-centered cubic (FCC), and the inclination angle fluctuates between 0° and 90°, allowing the bearing requirements for engineering structures to be satisfied. However, the unexpected geometric and process constraints have a high probability of causing a seemingly successful drawing to become a failed printing operation. Even if lattice cells are printed successfully, there are many manufacturing defects, which result in a significant deterioration in the mechanical properties [11]. As a means of solving these problems, the topology optimization method can partially deal with the minimum size and inclination angle constraints. However, it also faces several problems, such as three-dimensional (3D) space optimization and computational efficacy issues [12,13]. In addition, the mechanical properties are also obviously affected by the selected materials and the process parameters of 3D printing equipment [14,15]. The optimization of selected materials and process parameters of 3D printing equipment can also improve the mechanical properties of products [16,17]. The design, fabrication, and performance of selective laser melting (SLM) lattice structures have been introduced. The data are summarized to analyze the reported mechanical performance of SLM lattice structures and provide insight into the bounds of their technical capabilities [18,19]. However, this kind of work belongs to the field of optimization design for 3D printing equipment. Ordinary engineers mainly focus on the structural design of products rather than the design of 3D printing equipment. Therefore, for ordinary engineers, finding new structural design methods is the main way to obtain high-performance products. In fact, developing a new lattice product design method is a feasible way to improve the performance of SLM lattice structures.

In practice, the local reinforcement operation used to modify an existing geometric model, such as changing the angle of inclination and the radius of the rods, significantly increases the amount of effort required and component weight. Therefore, it would be very important if the printable lattice structure could be redesigned to enhance the lattice structure’s mechanical properties without changing the cell configuration. In the past five years, there has been a lot of interest in improving the performance of lattices by using hybrid methods in which the cell configurations of the lattice do not have to be changed. For instance, Pham et al. [20,21,22] designed a damage-tolerant hybrid lattice structure inspired by the crystal microstructure. They also revealed the underlying mechanisms responsible for strengthening mechanical properties. However, the inclination angles of hybrid lattice structures have a wide distribution range, which makes it difficult to ensure successful printing [23]. Designing a self-supporting lattice structure is an effective approach [24]. However, it is also limited by process constraints and cell configuration.

In general, under the current state of technology, manufacturing defects are difficult to avoid. The traditional method of strengthening the cell size will lead to an increase in the overall weight of the structure. Additionally, the topology optimization method has problems such as 3D space optimization and computational efficiency. This makes us have to find a new design method for structural design to meet the needs of lightweight, fewer machining defects, and easy manufacturing. Animals and plants exhibit excellent mechanical performances, and understanding their structural features can aid in the design of lattice structures with superior performances [25,26,27,28]. Several studies have been conducted on bionic designs, which include gradient and multi-layer designs. These strategies can easily produce false geometric designs. For example, the minimum size or tilt angle of the rod is too small, which leads to the failure of the printing of the bionic design components. The fastest-growing Bambusa emeiensis, with excellent compression and shear properties [25,26], is primarily composed of periodic bamboo joints and thinned tubes at the macroscopic level (see Figure 2a–c) [26,27]. At the mesoscopic level, the cross-section of the tube wall has a gradient distribution of fibers from the inside to the outside. Without changing the lattice cell configuration, this study introduced a macroscale bamboo joint to establish a cell partition layer and a microscale gradient distribution tube wall to appropriately set the variable rod radius in the lattice structure (see Figure 2d,e) to enhance the tube’s mechanical properties.

## 2. Experiment of Bambusa Bionic Lattice Structure

### 2.1. Bambusa Bionic Design for Lattice Structure

In this section, the Bambusa bionic lattice structure composed of joint and variable cell partitions is proposed, as shown in Figure 2e. The joint partition is composed of BCC lattice cells with a uniform rod, and the variable-cell partition is composed of a gradient rod, as shown in Figure 3. This can be easily printed using the selective laser melting (SLM) process. The uniform rod diameter is denoted as D. To describe this bionic design more accurately, we used the following bamboo bionic parameter *β* to analyze and characterize the joint and variable cell partitions:(1)$$\beta =\frac{V_{uni}}{V_{var}}$$
(2)$$V_{uni}=\sqrt{3}L_{u}\pi {\left(\frac{D_{max}}{2}\right)}^{2}$$
(3)$$V_{var}=2\times 0.4D_{max}\pi (\frac{D_{max}}{2}{)}^{2}+\int_{-(\frac{\sqrt{3}}{2}Lu-0.4D_{max})}^{\left(\frac{\sqrt{3}}{2}Lu-0.4D_{max}\right)}\pi f^{2}(x)dx$$
where $V_{uni}$ is the volume of the joint partition and $V_{var}$ is the volume of the variable partition. β represents the volume ratio of the joint partition to the variable partition when the mass is constant. D is the uniform rod diameter. $D_{max}$ is the maximum diameter of the section at the two ends of the variable rod containing a gradient cross-section, and $D_{\min}$ is the minimum diameter at the intermediate regions of the variable rod. *Lu* is the length of the two plotted lattice cells displayed in Figure 3, which were named the original lattice (O-lattice) and bamboo lattice (bamboo lattice). $f\left(x\right)$ is the curve function of the bar section.

When changing the uniform and variable cell geometry dimensions, *β* varies widely, as shown in Figure 3c. All the geometric dimensions are listed in Table 1.

### 2.2. Sample Preparation and Experimental Procedure

The designed O- and Bambusa-lattice samples were fabricated by selective laser melting (SLM). The relevant 316L stainless steel process parameters are listed in Table 2. To characterize the microstructure of the powder, scanning electron microscopy (SEM) investigations were also conducted on a Quanta FEG250 microscope. Figure 4a shows the SEM image of the 316L stainless-steel powder, with a particle diameter ranging between 40 and 55 µm, where good sphericity could be observed.

The lattice samples are shown in Figure 4b,c. Quasi-static compression tests for the lattice samples were performed, and their mechanical properties and deformation processes were analyzed. All the samples were compressed on an Instron universal testing machine equipped with a 30-kN load cell. The constant compressive strain rate was 10^−3^/s. Two repeated experiments were carried out to ensure the accuracy of the experimental results. A video camera with a framerate of 30 frames per second was utilized to record the deformation processes. Table 2 lists the as-designed and as-fabricated masses of all the desired lattice patterns, and the maximum error was only 3.67%. Meanwhile, two uniaxial tensile 316L stainless steel samples were also fabricated to obtain the mechanical properties. The average results were then utilized as inputs into the finite element (FE) model for simulations, as discussed in Section 2.3. The average elastic modulus, initial yield strength, and ultimate strength were determined to be 93 GPa, 545 MPa, and 1240 MPa, respectively, based on the experimental results. Detailed experimental processes can also be found in our previous work [29].

### 2.3. Finite Element Analysis

A finite element analysis was performed to investigate the compression deformation and mechanical properties of the Bambusa-lattice using the ABAQUS software. As shown in Figure 5, the model contained two rigid bodies and lattice cells meshed with C3D10M solid elements. To determine the appropriate element size, mesh convergence analysis was also performed, and an element size of 0.20 mm was selected to ensure the calculation’s efficiency. Two rigid bodies were coupled with two reference points. The upper body carried the load, while the lower body was fixed. A compression velocity of 1.0 mm/s was applied for the quasi-static loading conditions. General contact with a penalty friction coefficient of 0.1 was chosen between the lattice cells and rigid bodies, whereas automatic node-to-surface contact was also adopted between the lattice cells. The elastic–plastic model was used to simulate the SLM 316L material. The detailed properties of the parent material were tested and are presented in Section 2.2. Detailed material parameters for the numerical simulation are obtained from the uniaxial tensile tests on the samples, which are provided in Table 3.

As shown in Figure 6a, The uniaxial tensile tests are carried out using the testing machine which is made Shenzhen Labsans Testing Machine Co., Ltd (Shenzhen, China). The nominal stress–strain data of the O- and Bambusa-lattice structures are shown in Figure 6b. The mechanical properties of the two experiments exhibited good consistency. As shown in the figure, the curves could be classified into three typical stages [1,27]: elastic, plateau, and densification stages, in which the crushing stress increased sharply despite the small change in the compressive strain. Moreover, it was clear that the mechanical performance of the Bambusa-lattice was superior to those of the O-lattice.

Figure 7a,b present a comparison of the stress–strain curves obtained from the simulations and experiments. The results of the simulations agreed well with those of the experiments. Moreover, the compressive modulus E, initial yield strength $\sigma_{\mathrm{ys}}$, and plateau stress $\sigma_{p}$ are all summarized in Table 4. The compressive modulus was the slope of the curve before yield deformation. The plateau stress and SEA values were calculated according to previously published methods [24,27]. A maximum error of 5.37% was observed in the compressive modulus of the O-lattice. For the Bambusa-lattice, the maximum error was only 4.84%, and the other errors were very small. These comparisons further demonstrate the accuracy of the FE model.

Figure 7c,d presents a comparison of the data obtained using the simulated deformation processes with the experimental data. For the O-lattice, the “X” shear band deformation breakthrough of the entire structure can be observed at a compressive strain ε=0.2 in addition to the energy of the entire structure absorbed through plastic deformation, where the crushing stress remained mostly unchanged. It should be noted that the shear deformation band limited the loading capacity [1,20,21,22,30]. Compared with the O-lattice, the Bambusa-lattice exhibited different deformation characteristics. When the compressive strain ε=0.2, the “X” shear band is interrupted by the joint partition, and more localized shear deformation is located in the variable partition. This delayed the yield process and plastic deformation evolution in all the lattice cells. Later in Section 3.2, we can see how β interrupts the shear band (Figure 9). As listed in Table 4, the compressive modulus and initial yield strength increased by 21.36% and 18.88%, respectively. In other words, the joint partition was more susceptible to changing the shear band evolution and improving the mechanical properties. Additionally, from the experimental tests, the maximum error of two repetitions is 2.71%; this shows that the Bambusa bionic design method meets the requirements of the printing process, and the product quality is high.

## 3. Results and Discussion

### 3.1. Parametric Analysis for Mechanical Properties of Bamboo-Lattice Structure

In practical applications, compression and shear performances are key indices of lattice structures. Accordingly, the compression and shear properties of three bamboo lattice structures with different *β* values were analyzed and calculated. The geometric models are shown in Figure 8, and the material properties, constitutive model, and boundary conditions are presented in Section 2.3.

### 3.2. Compression Deformation Characteristics

Figure 9 illustrates the deformation process of the bamboo lattice structure. When compressed to a strain of 0.2, the localized deformation phenomenon in the joint partition impeded the shear band formation in the lattice structure. When the parameter β was equal to 0.6, the cell rod in the joint partition was thinner than that of the variable partition, as represented by the red boxes in Figure 9a. As a result, it was more susceptible to yielding, which generated a local compression region, and the shear band was completely separated. However, when β is equal to 1.5, there is no local compression region. By comparison, it can be concluded that the joint partition could interrupt the shear band evolution. Moreover, as shown in Figure 9b, as the compressive strain increased, the plastic deformation of the joint partition increased gradually and moved similarly to that of a rigid body with the upper and lower variable partitions compressing in the variable partition. In other words, the cell rods in the joint partition are first compacted in the compression process. The crushing stress increased significantly because of the larger cell rods in the upper and lower variable partitions. Similarly, when *β* = 1.0 and 1.5, plastic deformation occurred earlier in the variable partition with the thinner cell rod. It was also found that shear deformation band evolution was blocked. As the strain increased, the plastic deformation in the joint partition increased very slowly, and the crushing stress increased gradually compared with that of the Bambusa-lattice with *β* = 0.6, as shown in Figure 9c. When *β* > 1, the upper and lower rods are first compacted; furthermore, when *β* = 1 and 1.5, the deformation of rods in the joint partition is almost consistent, as seen in Figure 9b. Therefore, the compression characteristics of the Bamboo-lattice structure when *β* = 1 and 1.5 are different from those when *β* = 0.6. It can guide us in selecting the value of *β* in the actual design.

### 3.3. Shear Deformation Characteristics

The shear deformation and stress–strain curves are shown in Figure 10. The bamboo lattice with *β* = 0.6 had a higher shear modulus than the O-lattice structure. In Figure 10a,b, it can be observed that the stress distribution of the O-lattice structure was concentrated near the “X” shear band and that the von Mises stresses in other areas were significantly low. However, the stress distribution was directly changed in the bamboo lattice structure by the joint partition. For example, when parameter *β* was equal to 0.6, the stress distributions of the bamboo lattice structure were more uniform than those of the O-lattice structure. When parameter *β* was equal to 1.5, areas of stress concentration were found in the variable cell. In Figure 10c, it can be observed that parameter *β* altered the shear stress in the bamboo lattice structure. The relevant mechanical properties are listed in Table 5. Compared with the traditional method, the elasticity modulus of the product based on the bamboo bionic method can be increased by 1.51 times (β=1.5). The *β* parameter significantly affected the mechanical properties, particularly the shear properties.

Based on the above analysis, it was concluded that the Bambusa bionic design strategy can be used to enhance the mechanical properties of the lattice structure without changing the cell configuration or increasing the mass density. Moreover, a multi-functional satellite cabin can be designed with a thermal control channel and load-bearing characteristics (see Figure 11) to satisfy the in-plane shear and out-of-plane compression design requirements. The dividing line between different areas is the blue line in Figure 11a.

## 4. Conclusions

In this paper, a Bambusa bionic design strategy was innovated to improve the mechanical properties of lattice structures. The compression and shear deformation of the Bambusa-lattice structure were compared with those of the original lattice structure. Based on the experimental and simulation results, it was found that our design could significantly improve the mechanical properties without adding mass or changing the cell configuration. This comprehensively adjusted the evolution of the shear deformation band. Compared with the traditional method, the elasticity modulus of the product based on the bamboo bionic method can be increased by 1.51 times (β=1.5). By changing the Bambusa bionic parameter β, Young’s modulus and shear modulus could be easily enhanced. From the point of view of improving the mechanical performances of AM lattice structures, this design strategy can make up for the performance degradation due to process constraints and meet the design requirements.

In this study, successful attempts were made to enhance lattice structures, and the static properties of bionic bamboo-designed metal metamaterials were investigated. Nevertheless, no further optimization design to increase the mechanical properties was performed. Meanwhile, the vibration characteristics were not studied. Therefore, we will study this work step by step in the future.

## Figures and Tables

**Figure 1 materials-16-02545-f001:**
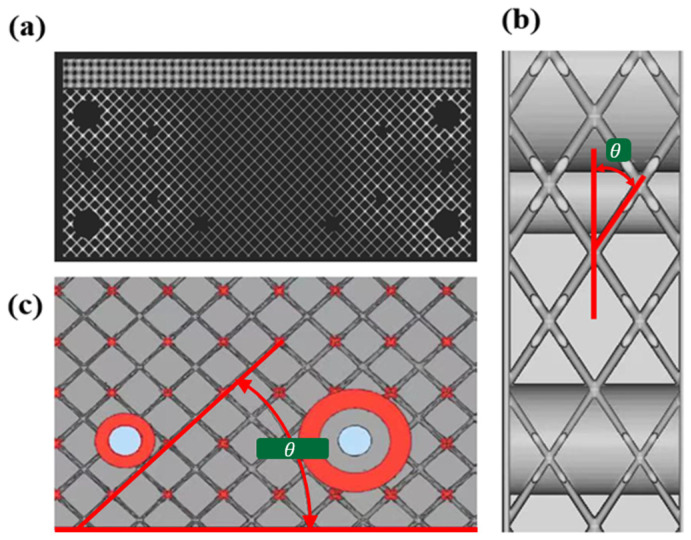
Satellite cabin structure with lattice cells: (**a**) various cell configurations; and (**b**,**c**) cell inclination angle in the printing direction.

**Figure 2 materials-16-02545-f002:**
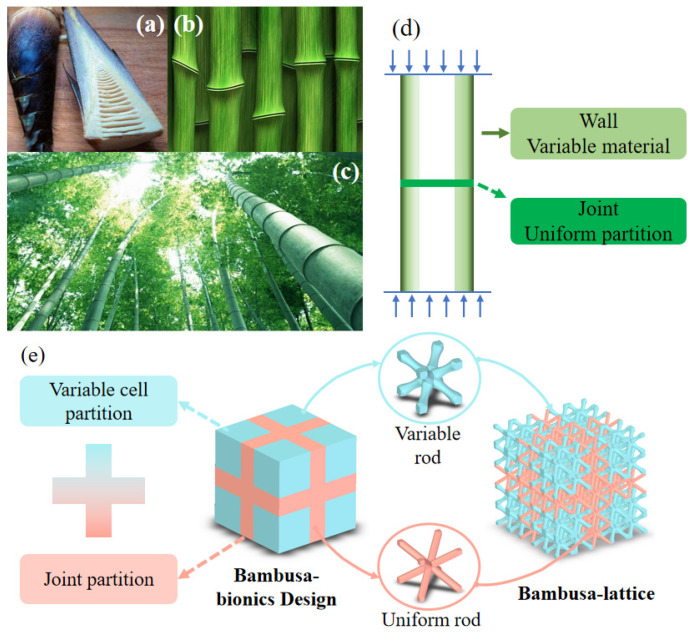
Bambusa-lattice structure inspired by bambusa joints and gradient tube walls: (**a**–**d**) Bambusa joint and tube structures; and (**e**) Bambusa bionic design.

**Figure 3 materials-16-02545-f003:**
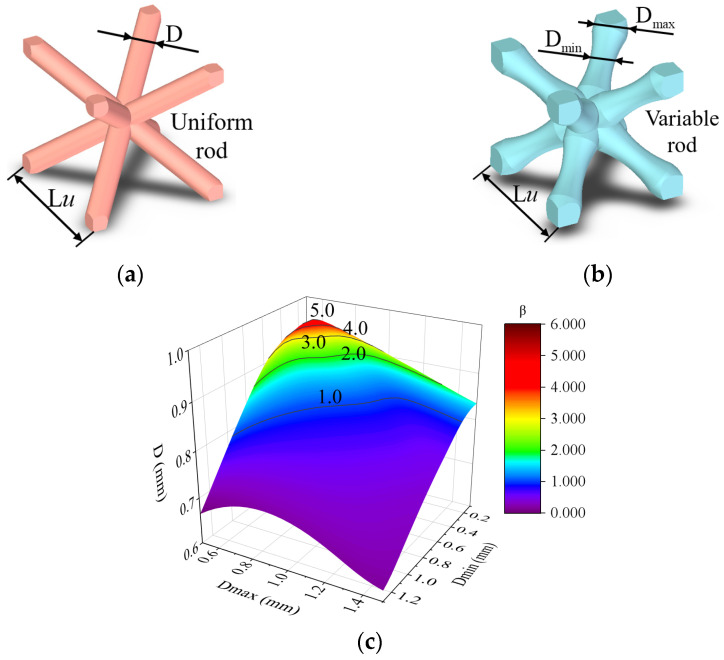
Geometries of uniform and variable cells: (**a**) uniform rod, (**b**) variable rod, and (**c**) bamboo bionic parameter *β*.

**Figure 4 materials-16-02545-f004:**
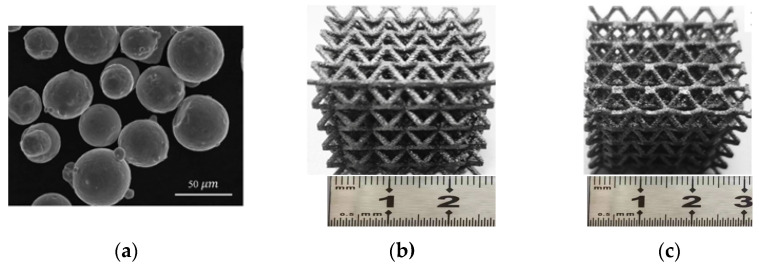
(**a**) 316L stainless steel powder, (**b**) O-lattice, and (**c**) Bambusa-lattice.

**Figure 5 materials-16-02545-f005:**
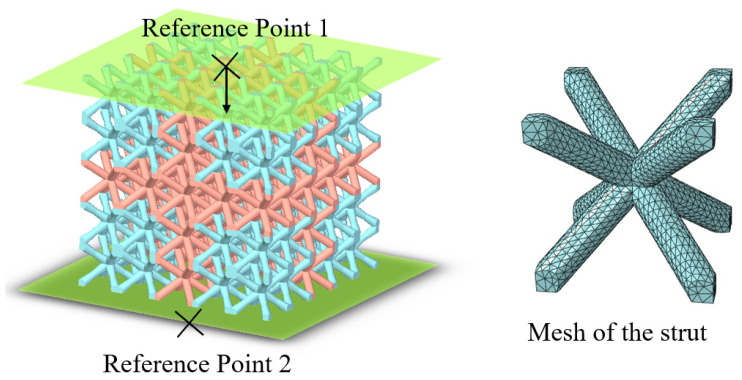
Finite element model of the Bambusa-lattice.

**Figure 6 materials-16-02545-f006:**
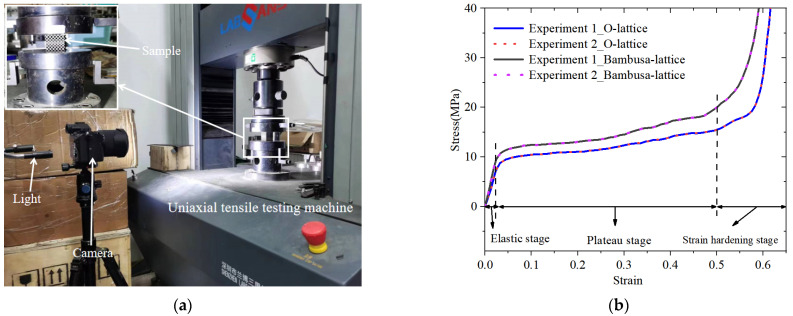
The experimental testing. (**a**) Uniaxial tensile test, (**b**) Quasi-static stress–strain curves of the O-lattice and Bambusa-lattice samples.

**Figure 7 materials-16-02545-f007:**
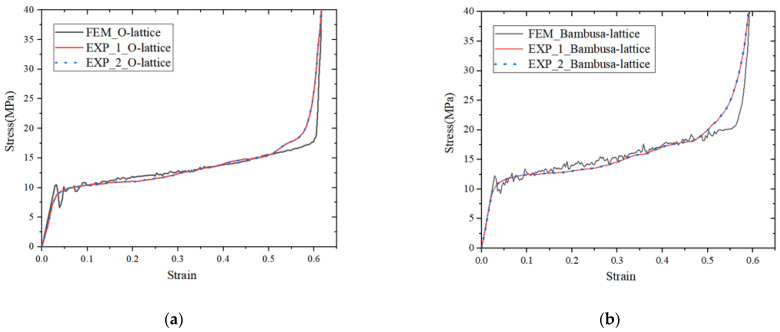
Comparisons of the stress–strain curves and deformation processes from experiments and simulations: (**a**,**c**) O-lattice samples and (**b**,**d**) Bambusa-lattice samples.

**Figure 8 materials-16-02545-f008:**
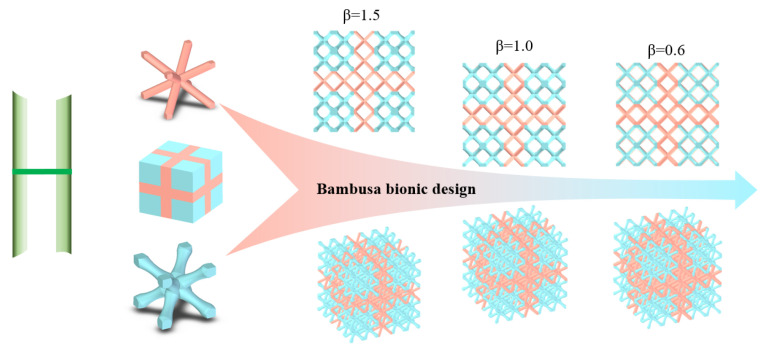
Three Bambusa-lattice structures.

**Figure 9 materials-16-02545-f009:**
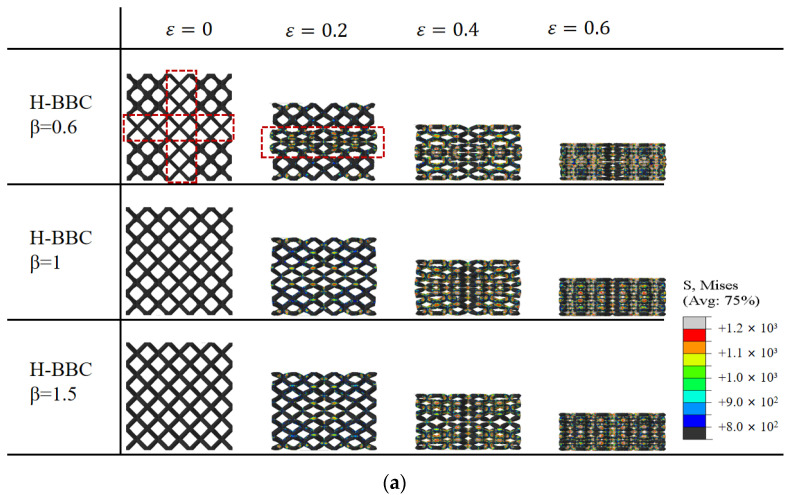
Compression characteristics of Bamboo-lattice structure: (**a**) deformation process in cross-sectional view; (**b**) middle unit cell in joint partition; and (**c**) stress–strain curve.

**Figure 10 materials-16-02545-f010:**
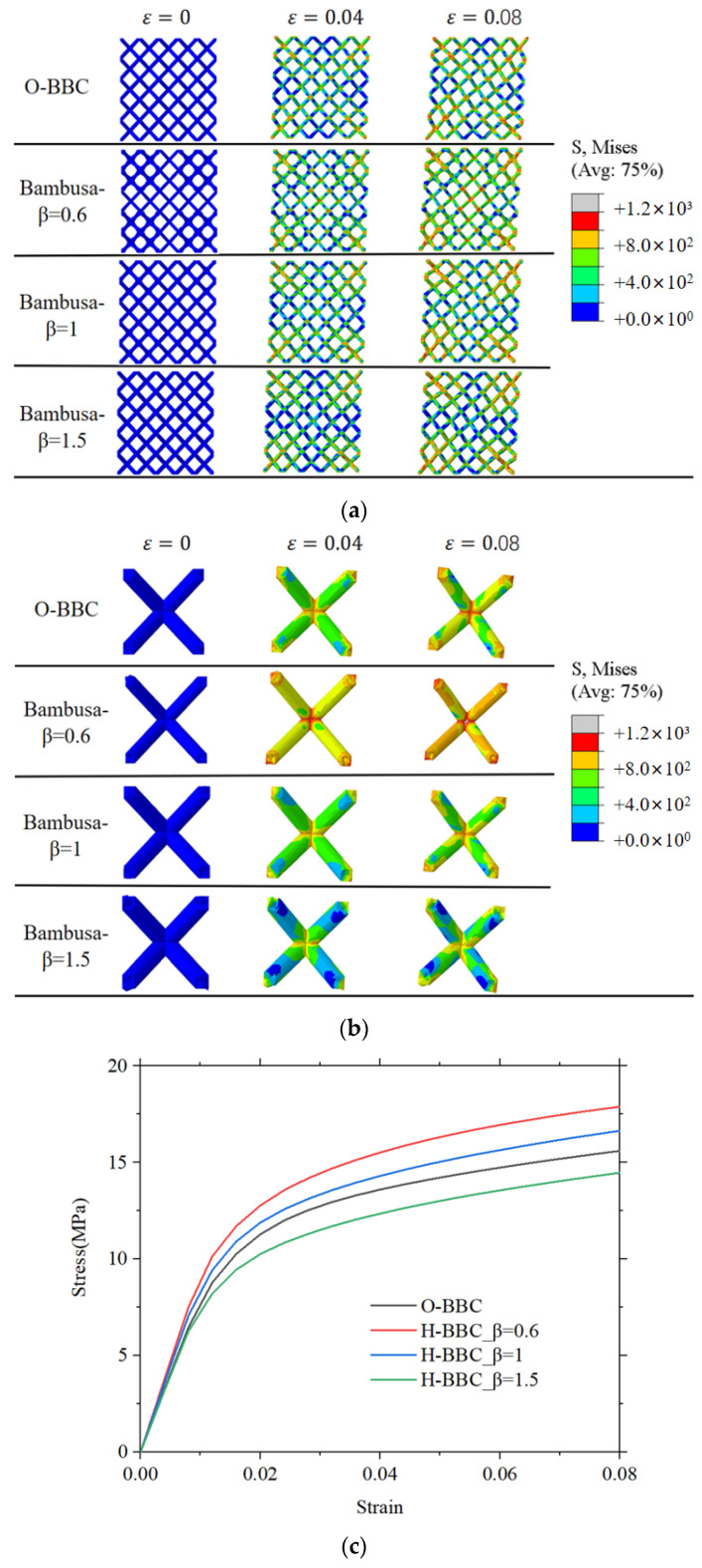
Compression and shear characteristics of Bamboo-lattice structures: (**a**) deformation process; (**b**) middle unit cell in joint partition; and (**c**) stress–strain curves.

**Figure 11 materials-16-02545-f011:**
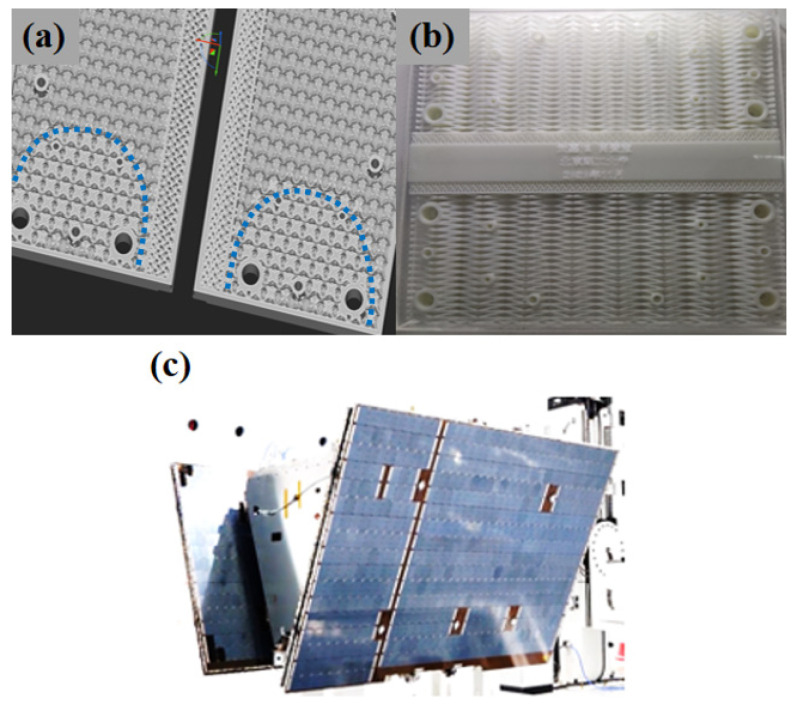
Additive-manufacturing satellite cabin filled with Bambusa-lattice cells: (**a**) load-bearing panel filled with Bambusa-lattice cells; (**b**) multifunctional panel with a thermal control channel and Bambusa-lattice cells; and (**c**) satellite cabin.

**Table 1 materials-16-02545-t001:** Design dimensions of the lattice structure.

Sample	Relative Density	Cell Size (mm)	Dimensions (mm^3^)	Diameter (mm)
O-lattice	0.1634	*Lu*: 5	25 × 25 × 25	*D*: 0.952
Bambusa-lattice (*β* = 1.0)	0.1634	*Lu*: 5	25 × 25 × 25	*D_max_*: 1.130 *D_min_*: 0.794

**Table 2 materials-16-02545-t002:** Printing process parameters and masses of the lattice samples.

Laser Power	Laser Exposure Time	Scanning Speed	Layer Thickness	Hatch Spacing
400 W	200 μs	0.15 m/s	30 μm	120 μm
	Designed mass (g)	Measured mass (g)	Average (g)	Error (%)
O-lattice	19.91	19.16 19.20	19.18	3.67
Bambusa-lattice	19.91	19.44 19.42	19.43	2.41

**Table 3 materials-16-02545-t003:** Material parameters of the parent material 316 L stainless steel [29].

Sample	Density (g/cm^3^)	Elastic Modulus (GPa)	Initial Yield Stress (MPa)	Ultimate Stress (MPa)	Ultimate Strain
1	-	95	516	927	0.2365
2	-	91	509	937	0.2374
Average	7.96	93	512.5	932	0.237

**Table 4 materials-16-02545-t004:** Mechanical properties of the O-lattice and Bambusa-lattice structures.

Lattice Configuration			E (MPa)	σ_ys_ (MPa)	σ_p_ (MPa)
O-lattice	Experiment	1st test 2nd test	426.85 424.38	8.912 8.868	10.575 10.870
		Average	425.62	8.89	10.72
		Exp Error(%)	0.58	0.49	2.71
	Simulation		402.77	9.147	11.188
	Error (%)		5.37	2.89	4.34
Bambusa-lattice	Experiment	1st test 2nd test	510.68 522.40	10.528 10.608	12.478 12.450
		Average	516.54	10.568	12.464
		Exp Error(%)	2.244	0.754	0.225
	Simulation		541.53	10.354	12.491
	Error (%)		4.84	2.02	0.22

**Table 5 materials-16-02545-t005:** Mechanical properties of the Bambusa-lattice structures.

β	E (MPa)	G (MPa)	σ_ys_ (MPa)	σ_m_ (MPa)
0.6	402.77	980	9.147	11.188
1.0	541.53	900	10.354	12.491
1.5	609.83	780	5.713	11.941
O-lattice	402.77	820	9.147	11.188

## Data Availability

The data can be got in this paper.

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
