# Peer review of "Enhancing Mechanical Properties of 3D Printing Metallic Lattice Structure Inspired by Bambusa Emeiensis"

_materials, 2023, doi:10.3390/ma16072545_

Round 1
Reviewer 1 Report
The manuscript discuss the challenges involved in the design and fabrication of lattice structures using additive manufacturing (AM) technology, particularly when it comes to metallic components filled with lattice cells. The manuscript also introduce a bionic design inspired by Bambusa, which consists of a macroscale bamboo joint to establish a cell partition layer and a microscale gradient distribution tube wall to appropriately set the variable rod radius in the lattice structure, with the aim of enhancing the mechanical properties of the lattice structure. The design is analyzed and characterized using the bamboo bionic parameter β, and the samples are fabricated using selective laser melting (SLM). Quasi-static compression tests are performed to determine the mechanical properties and deformation processes of the lattice samples.
The manuscript is generally well-written and informative. However, some potential criticisms include of the manuscript:
1. The introduction cites several sources but does not provide a comprehensive literature review of the current state of research in the field of AM lattice structures and their mechanical properties.
2. The manuscript does not provide a clear conclusion or summary of the research findings.
3. Although the authors claim that these structures are more efficient and mechanically strong, defects in the internal structure can lead to failure more quickly. So it should be exported to an SEM image for the internal defect.
Reviewer 2 Report
This work aims to study bamboo inspired structures but the structures the authors came up with (just nodes with varying rod diameter) seems to be a bit of a stretch to compare to bamboo structures. Bamboo structures are famous for their hierarchical multiscale structures. Also, language and writing in general needs to be revised. Quite a few awkward sentences with confusing sentence structures.
Line 83 clarify meaning of sentence. What do you mean by false designs?
Line 100, full description of acronym SLM should have been introduced.
Lines 106~111, description of o-lattice and bamboo lattice is confusing. Figure 2 has bamboo lattice composed of both the o-lattice and the variable rod unit cells. Also hard to verify the difference between the lattice structures in figure 4 optical images.
Figure 4 needs scale bars for figure b and c
Line 179: when the compressive strain “was” or “is”
Figure 7 needs explanation of the red arrows representing the shear band. Perhaps more simulation data should be included in supporting materials. Just from the figure alone, it’s hard to see how the red arrows correctly represents the shear bands.
Line 183: how does the joint partition interrupt the shear band evolution? It’s hard to see from the figure where the joint partition is located.
Line 202: what do you mean by moving “almost rigidly”?
Figure 9(c): how do you explain the similar behavior for beta=1 and 1.5?
Fig 11. Show more clearly how these structures have bambusa-lattice cells. Hard to see from the images. Perhaps a zoom in fig.
Reviewer 3 Report
In this paper, the mechanical properties of an AM lattice structures are improved using a Bambusa bionic design. The topic is relevant, the results are good but some clarifications are needed.
General comment
- Please clarify all the parameters and formulas you use.
- Please indicate the units of measurement of the parameters in all figures and tables.
- What is your opinion on the buckling behavior in this problem?
Abstract
- Please stress the results of the FEM analyses and experimental tests.
Introduction
- The introduction should report the latest analyses and experimental tests conducted on similar problems. Stress the novelty of your study.
Reviewer 4 Report
There are some weaknesses through the manuscript which need improvement. Therefore, the submitted manuscript cannot be accepted for publication in this form, but it has a chance of acceptance after a major revision. My comments and suggestions are as follows:
1- Abstract gives information on the main feature of the performed study, but ac couple of sentences about the obtained results must be added.
2- Authors must clarify necessity of the performed research. Research questions, aims and objectives of the study must be clearly mentioned in introduction.
3- The literature study must be enriched. In this respect, authors must read and refer to the following papers: (a) https://doi.org/10.1038/s41598-022-05005-4 (b) and other research works.
4- Authors must explain the limitations of their study.
5- The group (bunch) citation must be avoided. For example: [6-11] in introduction. Also, all figures must be illustrated in a high quality. For example, Fig. 5
6- Why this particular material and printing method are considered in this study.
7- Since it is experimental study, at least a real figure is required to show specimen under test condition.
8- Why this particular strain rate is used for the experiment? Is there any standard? Must be explained in detail.
9- The main reference of each formula must be cited. Moreover, each parameters in equations must be introduced. Appropriate references for the values presented in Tables are required.
10- Authors claimed that “video camera with a framerate of 30 frames per second was utilized to record the deformation processes”. The captured images must be added to the manuscript.
11- Input parameters of FE must be summarized in a table.
12-Standard deviation is the presented curves must be discussed. In addition, error in calculation must be considered and discussed.
13- In its language layer, the manuscript should be considered for English language editing. There are sentences which have to be rewritten.
14- The conclusion must be more than just a summary of the manuscript. List of references must be updated based on the proposed papers. Please provide all changes by red color in the revised version.
Round 2
Reviewer 1 Report
The authors made the corrections and additions that I mentioned in the previous review.
Reviewer 2 Report
Although writing can be further improved, it is acceptable in its current format.
Reviewer 4 Report
The paper has been improved and corresponding modifications have been conducted. In my opinion, the current version can be considered for publication.